# Understanding Carbohydrate Metabolism and Insulin Resistance in Acute Intermittent Porphyria

**DOI:** 10.3390/ijms24010051

**Published:** 2022-12-20

**Authors:** Isabel Solares, Daniel Jericó, Karol M. Córdoba, Montserrat Morales-Conejo, Javier Ena, Rafael Enríquez de Salamanca, Antonio Fontanellas

**Affiliations:** 1Reference Center for Inherited Metabolic Disease-MetabERN, Department of Internal Medicine, University Hospital 12 de Octubre, UCM, 28041 Madrid, Spain; 2Hepatology Program, CIMA Universidad de Navarra, 31008 Pamplona, Spain; 3Navarra Institute for Health Research (IdiSNA), 31008 Pamplona, Spain; 4Grupo de Enfermedades Mitocondriales y Neuromusculares, Instituto de Investigación Hospital 12 de Octubre (i+12), Centro de Investigación Biomédica en Red de Enfermedades Raras (CIBERER), Instituto de Salud Carlos III, 28029 Madrid, Spain; 5Department of Internal Medicine, Marina Baixa Hospital, 03570 Villajoyosa, Spain; 6Centro de Investigación Biomédica en Red de Enfermedades Hepáticas y Digestivas (CIBERehd), Instituto de Salud Carlos III, 28029 Madrid, Spain

**Keywords:** hepatic porphyrias, acute intermittent porphyria, fasting, glucose homeostasis, insulin resistance, mitochondrial function and biogenesis

## Abstract

Porphobilinogen deaminase (PBGD) haploinsufficiency (acute intermittent porphyria, AIP) is characterized by neurovisceral attacks associated with high production, accumulation and urinary excretion of heme precursors, δ-aminolevulinic acid (ALA) and porphobilinogen (PBG). The estimated clinical penetrance for AIP is extremely low (<1%), therefore it is likely that other factors may play an important role in the predisposition to developing attacks. Fasting is a known triggering factor. Given the increased prevalence of insulin resistance in patients and the large urinary loss of succinyl-CoA to produce ALA and PBG, we explore the impact of reduced availability of energy metabolites in the severity of AIP pathophysiology. Classic studies found clinical improvement in patients affected by AIP associated with the administration of glucose and concomitant insulin secretion, or after hyperinsulinemia associated with diabetes. Molecular studies have confirmed that glucose and insulin administration induces a repressive effect on hepatic ALA Synthase, the first and regulatory step of the heme pathway. More recently, the insulin-mimicking α-lipoic acid has been shown to improve glucose metabolism and mitochondrial dysfunction in a hepatocyte cell line transfected with interfering RNA targeting PBGD. In AIP mice, preventive treatment with an experimental fusion protein of insulin and apolipoprotein A-I improved the disease by promoting fat mobilization in adipose tissue, increasing the metabolite bioavailability for the TCA cycle and inducing mitochondrial biogenesis in the liver. In this review, we analyze the possible mechanisms underlying abnormal hepatocellular carbohydrate homeostasis in AIP.

## 1. Introduction

Acute intermittent porphyria (AIP, MIM 176000) is an autosomal dominant disease caused by a partial deficiency of the hepatic porphobilinogen deaminase (PBGD, EC 4.3.1.8), the third enzyme of the heme synthesis pathway. AIP is characterized by acute attacks of abdominal pain, nausea, vomiting and fatigue that can be triggered by endogenous or exogenous factors, such as a low carbohydrate diet or fasting, that up-regulate the expression of the first enzyme of the pathway, δ-aminolevulinate (ALA) synthase 1 (ALAS1, EC 2.3.1.27) [1,2]. This mechanism can cause an excessive accumulation of toxic substrates, ALA and porphobilinogen (PBG) [3,4,5].

Various studies have reflected a high prevalence of HMBS enzyme mutations, with figures of around 1 in 1700 individuals. Penetrance, however, is markedly low, less than 10% among families of AIP patients, and could be as low as <=1% depending on the degree of underdiagnosis [6]. Thus, not all carriers of a mutation in HMBS will go on to experience an acute attack of porphyria in the course of their lives, but the association with other causative or modifier genes, or other inducing factors will be necessary to trigger these attacks. Furthermore, a substantial number of asymptomatic patients maintain a high excretion of ALA and PBG, which suggests that the pathophysiology of acute events cannot only be associated with the accumulation of neurotoxic by-products. Homedan et al. [7] suggest that the removal of succinyl-CoA from the TCA, used to support the increased demand for heme synthesis in the context of an AIP attack, could have a profound, although reversible, impact on mitochondrial energy exchange. Several additional studies have addressed the metabolic aspect of acute porphyrias that could be associated with an energy misbalance due to sustained overproduction of heme-precursors in the liver [8]. Indeed, reduced serum accumulation of insulin-like growth factor 1 and transthyretin, associated with a state of under-nutrition and/or with hepatic inflammation due to the sustained accumulation of heme-precursors, have been proposed as biomarkers of morbidity/severity of the disease for the clinical follow-up of patients with AIP [9].

A balanced diet of proteins and fats and a carbohydrate intake of 45–60% of total energy intake is recommended in patients with acute porphyrias [10,11]. Carbohydrate loading is especially important when patients begin to have emerging porphyria symptoms.

The effect of diet on experimental porphyria was known even before the biochemical basis of porphyria was discovered. In 1961, Rose et al. [12] observed that the administration of carbohydrates reduced the urinary excretion of PBG in a murine model of porphyria induced by 2-allyl-2-isopropyl-acetamide (AIA), a porphyrinogenic drug. This finding was initially interpreted as a direct interaction between AIA and carbohydrates. However, in 1964, Tschudy [13] demonstrated that the administration of carbohydrates inhibits the induction of ALAS1. This finding explained why fasted animals had a greater overproduction of heme precursors than non-fasted animals in experimental models of porphyria [14]. Subsequently, it was found that the excretion of ALA and PBG is also modified by the amount of carbohydrates ingested in humans [15]. Since then, carbohydrate overload has been administered as a treatment during acute attacks of porphyria, although the exact mechanism by which carbohydrates exerted this effect was not described until 2005 [16].

## 2. Carbohydrate Metabolism in AIP

Glucose is an essential human nutrient and acts as the main energy supply. The liver has a primary role in glycemic control. It regulates the balance between glucose storage, through glycogenogenesis and its release, through glycogenolysis and gluconeogenesis [17].

In response to low blood sugar and the resulting increase in glucagon, hepatocytes activate glycogen breakdown to release glucose into the bloodstream for uptake by other cells. The enzyme glycogen phosphorylase (GP, EC 2.4.1.1) plays a major role in the mobilization of glucose stored in tissues during glycogenolysis. This enzyme catalyzes the cleavage of glycogen to glucose-1-phosphate [18]. Insulin also inhibits this enzyme by promoting the phosphorylation of transcription factors and blocking its expression [19].

Gluconeogenesis is the process of glucose formation from non-glucidic substrates such as lactate and alanine. This metabolic pathway is essential for glucose production during prolonged fasting. A key step in gluconeogenesis is the formation of phosphoenolpyruvate from oxaloacetate, which is catalyzed by the enzyme phosphoenolpyruvate carboxykinase (PEPCK EC 4.1.1.32) [20]. The transcription of PEPCK is regulated by multiple factors, both dietary and hormonal: cyclic AMP, glucocorticoids and thyroid hormones increase PEPCK gene expression [21]. By contrast, insulin inhibits its transcription [22].

In 2005, Lelli et al. [23] described a blockade of the gluconeogenesis and glycogenolysis pathways in pharmacologically induced porphyric mice. This blockade seems to be a consequence of a decrease in hepatic expression of PEPCK and GP. This finding was corroborated by Collantes et al. [24], who described deficient induction of glycogenolysis in fasted AIP mice when compared with wild-type (WT) mice. By contrast, AIP mice activated the ketogenic pathway. Those PBGD-deficient mice also showed impaired glucose on a tolerance test, as well as serum hyperinsulinemia compared with the WT group. Notably, both parameters were normalized when hepatic PBGD deficiency was corrected through liver gene therapy. These data suggest that insulin resistance is associated with hepatic PBGD deficiency. Other authors [25,26] had previously suggested that the increase in the insulin/glucagon ratio is responsible for the abnormalities in oral tolerance to glucose in patients with porphyria

## 3. Role of Insulin in the Heme Synthesis Pathway

The counterregulatory hormones insulin and glucagon are responsible for the regulation of ALAS1 during periods of fasting and feeding through coactivator 1-alpha of peroxisome proliferator-activated receptor gamma (PGC-1α) [16] (Figure 1). PGC-1α is a main transcriptional coactivator for the control of mitochondrial biogenesis and hepatic gluconeogenesis [27]. This cofactor increases hepatic ALAS1 transcription through the interactions of the ALAS1 promoter with the transcription factor FOXO1 (forkhead box O1). In a dephosphorylated state, FOXO1 binds to PGC-1 and forms a transcriptional complex capable of inducing ALAS1 expression.

During fasting, glucagon induces PGC-1α through the stimulation of the cyclic AMP pathway. PGC-1α recruits transcription factors that bind the promoter of gluconeogenic genes, such as PEPCK, to stimulate beta-oxidation of fatty acids for energy supply during periods of prolonged fasting [27].

Therefore, the increase in serum glucagon stimulates the production of PGC-1α, which induces the synthesis of ALAS1. Through this mechanism, fasting causes activation of the heme biosynthetic pathway and can act as a trigger for an acute attack of porphyria [28] (Figure 1A).

By contrast, postprandial carbohydrate overload stimulates insulin secretion, which induces phosphorylation of FOXO1 and disrupts the transcriptional complex with PGC1a (Figure 1B). Therefore, insulin signaling through PI3K downregulates the overexpression of hepatic ALAS1 after glucose loading therapy [29]. Therefore, insulin resistance due to receptor deficiency or abnormal PI3K signaling in the liver may condition the therapeutic effect of glucose in acute porphyria.

Some groups have described a differential behavior in AIP depending on their insulinemia. Storjord et al. found that stable patients have higher insulin levels than those with symptoms [30]. Likewise, fasting insulinemia is lower in those patients who present a high level of urinary PBG [31].

## 4. Insulin Resistance in AIP

In 1949, Sterling et al. [32] were the first to report the association between the development of diabetes and porphyria. They described three cases of diabetes mellitus in a series of eight patients with AIP, although the impact of this relationship was unknown at that moment. Subsequently, Sixel-Dietrich [33] compared the results of an oral glucose tolerance test in an AIP patient during an acute crisis versus an asymptomatic carrier of the disease. He found that in the AIP patient, parallel to the decrease in neurotoxic precursors, there was a marked increase in serum glucose and insulin levels. These findings contrasted with a glycemic curve and an insulin concentration within the normal range presented by the asymptomatic control. Subsequently, several human case series [34,35,36] and experimental murine models [37] have confirmed the association between abnormal oral glucose tolerance tests and insulin resistance.

However, the occurrence of insulin resistance is not homogeneous among individuals with acute porphyria. Between Norwegian patients with AIP and controls, no significant differences were observed in insulinemia or the homeostasis model assessment (HOMA) index, which is a method for quantifying insulin resistance and beta-cell function; however, those patients with periodontitis showed an increased prevalence of insulin resistance [38].

In a recent observational case-control study, we compared the prevalence of insulin resistance in 44 Spanish patients with AIP and 55 age-, gender- and BMI-matched control volunteers. We found a significantly higher prevalence of insulin resistance and HOMA index in patients with stable disease than in patients with active porphyria (Figure 2) [39]. There were no significant differences in overweight, sedentary lifestyle, or metabolic syndrome between AIP patients and the control group; thus, finding significantly more insulin-resistant patients in the AIP group cannot be related to the factors characteristically associated with insulin resistance.

To our knowledge, no molecular mechanism has been described that could explain the pancreatic hypersecretion of insulin in some patients with AIP. The most widely accepted model of the pathogenesis of type 2 diabetes postulates that a high-fat diet leads to obesity and insulin resistance [40]. According to this theory, excess energy metabolites cause a reduction in insulin receptor signaling pancreatic β cells to release more insulin. However, the physiological mechanism of this hyperstimulation is unclear since it often occurs before hyperglycemia. Subsequently, pancreatic β cells become exhausted, leading to the development of type 2 diabetes. This hypothesis suggests that hyperinsulinemia is a compensation state for systemic insulin resistance [41]. Nevertheless, multiple studies have shown that insulin hypersecretion precedes obesity and insulin resistance, which is not consistent with the notion of hyperinsulinemia being simply an adaptive response to insulin resistance [42,43].

Shanik et al. [44] describe different situations in which insulin appears to be an essential quantitative contributor to insulin resistance, such as in patients with insulinomas. They conclude that hyperinsulinemia may result from and be a driver of insulin resistance. Thus, it is also possible that the hyperinsulinemia observed in patients with AIP is responsible for the subsequent development of insulin resistance. The exact mechanism by which patients with porphyria, especially those who are asymptomatic, present elevated insulinemia is still unknown. Matkovic et al. suggested that the increase in insulin levels in porphyria may be partly due to a decrease in its degradation [45]. The high production of reactive oxygen species in AIP promotes oxidative stress, which seems to interfere with the activity of glutathione-insulin transhydrogenase (EC 1.8.4.2), the enzyme responsible for insulin degradation [46]. The sustained increase in insulinemia in these patients may be responsible for the subsequent development of insulin resistance and the blockade of the gluconeogenesis and glycogenolysis pathways by causing the inhibition of PEPCK and GP, respectively (Figure 3).

Although the appearance of insulin resistance is not fully explained, our results suggest that the increase in insulinemia could be associated with the stabilization of porphyria, from both a clinical and biochemical point of view. This fact is supported by the known repressive effect of insulin on the transcription of PGC1α, a cofactor with the ability to repress ALAS1 transcription. Thus, insulin administration might be an innovative therapeutic tool for treating acute crises of porphyria

## 5. Insulin as a Therapeutic Weapon

One study already reported in 1970 showed how some patients experienced a clinical improvement when small amounts of insulin were administered together with carbohydrates during crises [47]. Experimental studies conducted by Handschin et al. [16] also found that the combination of glucose and insulin causes more potent inhibition of ALAS1 than administering glucose alone. On the other hand, although Oliveri et al. [48] did not find a reduction in the levels of PGC-1α with insulin administration, they observed that this treatment reduced the expression of ALAS1.

However, a possible drawback of using this drug could be the iatrogenic induction of hypoglycemia [39]. This could stimulate glucagon secretion; therefore, the effect obtained would be the induction of hepatic ALAS1. A proof-of-concept study with a recombinant fusion protein insulin coupled to apolipoprotein A-I (Ins-ApoAI) apolipoprotein A-I (Apo) confirmed a slow but prolonged mechanism of action, which diminishes the risk of hypoglycemia [49].

The Ins-ApoAI protein has a natural tropism for the liver [50,51] and a long biological half-life in the serum. We reported the repressive effect of the co-administration of glucose and Ins-ApoAI to counteract the direct induction of the hepatic ALAS1 gene modulated by extended fasting [39]. Furthermore, the administration of exogenous insulin tends to restore the expression of PEPCK and GP enzymes, regulators of the gluconeogenesis and glycogenolysis pathways, which suggests normalization of glucose homeostasis. In muscles, the recombinant Ins-ApoAI protein was also able to increase insulin sensitivity and enhance direct glucose uptake via AMP-activated protein kinase [52]. Finally, preventive treatment with insulin-ApoAI would also enable the beta-oxidation of fatty acids to be activated, as an important source of energy supply for the rest of the organs and of metabolites in the form of Acetyl-CoA that would be incorporated into the hepatocyte TCA cycle [39].

More recently, the administration of glucose and the insulin-mimicking α-lipoic [53] acid improved glucose metabolism and mitochondrial dysfunction in a hepatocyte cell line where the expression of PBGD was interfered with using a short interfering RNA [53]. The authors reported the restoration of the cross-talk between cytosolic glycolysis and mitochondrial respiration and, therefore, hepatocellular homeostasis.

However, we found that prophylactic administration of the recombinant Ins-ApoAI protein and glucose was not sufficient to achieve biochemical protection against a severe attack induced after recurrent phenobarbital administration [39]. The hepatic transcriptome of AIP mice revealed that phenobarbital also induced dysregulation of the gene involved in mitochondrial biogenesis (PGC1α) and oxidative phosphorylation, whose regulation closely depends on peroxisome proliferator-activated receptor gamma coactivator 1-alpha (PPARGC1A, encoding the PGC1-transcription factor) [28]. Notably, prophylactic administration of insulin-ApoAI was associated with behavioral improvements, probably related to the protective effect associated with the ApoAI-moiety promoting mitochondrial biogenesis at the hepatic level, as previously described by Song P. et al. [54]. Thus, further studies are needed to confirm whether new insulins or insulin-mimicking substances can improve insulin resistance, promoting glucose availability in other organs by restoring hepatic gluconeogenesis and glycogenolysis, and increasing mitochondrial dynamics and oxidative phosphorylation activity to increase energy production in hepatocytes.

## 6. Conclusions

Heme synthesis is fully integrated into the metabolism network because it uses glycine and succinyl-CoA from the TCA cycle. Hence, stimulation of heme biosynthesis during an acute attack requires TCA cycle replenishment, driving pyruvate and acetyl-CoA from the degradation of glucose, fatty acid β-oxidation, and ketogenic and glucogenic amino acids. All these data suggest the close integration of heme biosynthesis into carbohydrate, lipid and protein metabolism.

However, several studies have addressed the higher prevalence of insulin resistance and other alterations in glucose metabolism in patients with porphyria compared to healthy individuals. This alteration can be related to various causes such as the dysfunction of glutathione-insulin transhydrogenase and the chronic carbohydrate overload to which patients with AIP are exposed [39]. However, it seems that ALA itself can control glucose metabolism since several cohort studies have demonstrated the potential of ALA as a treatment for individuals with prediabetes and type-2 diabetes mellitus [55,56], and there is in vivo proof that ALA deficiency attenuates mitochondrial function and causes IGT and IR [57] even in porphyric mice subjected to the same diet as the wild type [24].

In addition, fasting secondary to nausea and vomiting and major losses of hepatic succinyl-CoA during an acute attack, as well as impaired glucose metabolism, reduce the availability of energy metabolites and could play a role in modulating the severity of porphyria attacks.

The intravenous administration of an experimental insulin-ApoAI protein or oral supplementation with a molecule that mimics insulin can improve glucose therapy by the repressive effect of insulin on the transcription of hepatic ALAS1, by increasing the energy supply to replenish the hepatocyte TCA cycle, as well as by enhancing mitochondrial respiration.

## Figures and Tables

**Figure 1 ijms-24-00051-f001:**
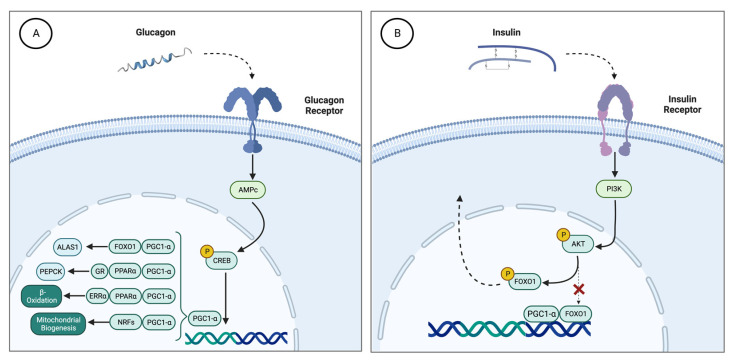
(**A**) Effect of glucagon on hepatic ALAS1 through PGC1a. During fasting periods, there is an increase in serum glucagon. This hormone stimulates the cAMP and CREB pathways, which activate the expression of the PGC-1 gene. Then, PGC-1α recruits different transcription factor binding promoters such as PEPCK (PGC1α, GR and PPARα) to induce gluconeogenesis, GP (FOXO, HNF4α and PGC1α) to enhance glycogenolysis, induces the expression of NRFs that increase mitochondrial DNA transcription and replication, and stimulates the heme biosynthetic pathway through the activation of ALAS1 (FOXO1 and PGC1α). (**B**) Effect of insulin on PGC1α. Carbohydrate intake produces an increase in serum insulin levels, with the consequent activation of the PI3K pathway. PI3K regulates the phosphorylation of AKT, which in turn phosphorylates FOXO1. Phosphorylation of FOXO1 disrupts the transcriptional complex it forms with PGC1α, such that FOXO1p translocates out of the nucleus and PGC1α loses its ability to bind to the ALAS1 transcriptional regulatory sequence. ALAS1, δ-aminolevulinate synthase 1; AMPc, cyclic adenosine monophosphate; FOXO1, forkhead box O1; NRFs, nuclear respiratory factor 1; PEPCK, phosphoenolpyruvate carboxykinase; GP, glycogen phosphorylase; PI3K, phosphatidylinositol-3 kinase; PGC1α, proliferator-activated receptor γ coactivator 1α.

**Figure 2 ijms-24-00051-f002:**
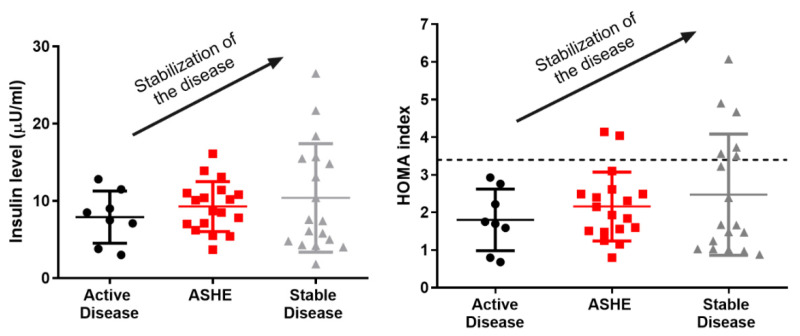
Plasma insulin quantification and HOMA index in porphyric patients. It is clear that, as the insulinemia and the HOMA of the patients rise, their disease stabilizes. AD: active disease, ASHE: asymptomatic high excreters, HOMA: homeostasis model assessment, SD: stable disease.

**Figure 3 ijms-24-00051-f003:**
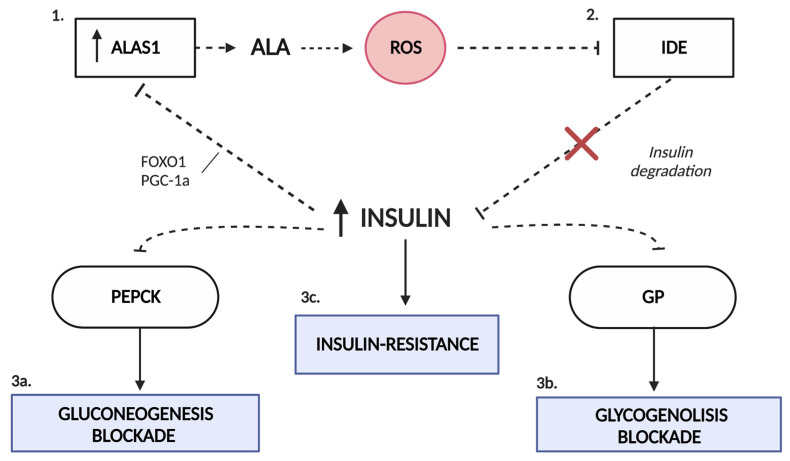
Interaction of carbohydrate metabolism and the heme biosynthetic pathway. The state of oxidative stress produced by the excess of ALA in porphyria (1) favors the dysfunction of the enzyme glutathione-insulin transhydrogenase, responsible for the degradation of insulin (2). Hyperinsulinemia alters the pathways of gluconeogenesis (3a) and glycogenolysis (3b) due to the inhibition that insulin exerts on PEPCK and GP enzymes. This increase in serum insulin favors the inhibition of ALAS1 by the disruption of the FOXO1-PGC1α complex (1). In addition, it is likely that it promotes the development of insulin resistance in porphyric patients (3c). ALAS1, δ-aminolevulinate synthase 1; FOXO1, forkhead box O1; GP, glycogen phosphorylase; IDE, insulin-degrading enzyme; PEPCK, phosphoenolpyruvate carboxykinase; PGC1α, proliferator-activated receptor γ coactivator 1α; ROS, reactive oxygen species.

## Data Availability

Not applicable.

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
