# Peer review of "Understanding Carbohydrate Metabolism and Insulin Resistance in Acute Intermittent Porphyria"

_ijms, 2022, doi:10.3390/ijms24010051_

Round 1

Reviewer 1 Report

This is an important publication that summarizes the current understanding of carbohydrate metabolism and insulin in Acute Intermittent Porphyria. This publication will be very useful to others in the field. 

There are several part of the publication that could benefit from clarification.

1.     Line 51: It is likely true that the penetrance is markedly low. However, the difference between the prevalence of pathogenic variants versus diagnosed cases could be from some combination of underdiagnosis and decreased penetrance. Underdiagnosis is also likely to be a large issue in the disease, because the time to diagnosis is around 10 years. The amount of underdiagnosis is unclear. Among families of AIP patients, the penetrance is around 10%. Please revise to allow for some possibility of underdiagnosis contributing to this <=1% figure. For instance, you could say the following: “Penetrance, however, is markedly low, less than 10% among families of AIP patients, and could be as low as <=1% depending on the degree of underdiagnosis.”

2.     Line 58. The way this sentence is written is confusing to me, and it sounds like succinyl-CoA from the TCA cycle is trying to produce ALA and PBG as an end goal, when ALA and PBG is an abnormal accumulation of a biochemical intermediate of heme synthesis. It would be clearer to say something like the following: “removal of succinyl-CoA from the TCA, used to support the increased demand for heme synthesis in the context of an AIP attack…” Something similar to this was said in lines 279-281 of the manuscript, and the meaning there is clearer to me.

3.     Line 107. This sentence should say “Those PBGD-deficient mice also showed impaired glucose on a tolerance test.” Please remove the unnecessary “an.” 

4.     Line 170. This is the first time HOMA is used in the article, and it is not defined. It is only defined in the legend of Figure 2, but it should be defined both places. I also think it would be helpful to give some description of what it is, as not everyone would be familiar with this test. One option would be to say the following: “Between Norwegian patients with AIP and controls, no significant differences were observed in insulinemia or the Homeostasis Model Assessment (HOMA) index, which is a method to quantify insulin resistance and beta-cell function; however, those…”

5.     Line 182. The legend of Figure 2 should say “It is clear that, as the insulinemia….,” not “it is clear how.” In the article, it is stated that the mechanism is unclear, so the “how” is not clear.

6.     Line 186-188. The authors say that there is a widely accepted model of type 2 diabetes pathogenesis, but there is no reference. It’s possible that article #40, which is listed in a few sentences, is meant to be the reference for this entire section. However, that is not very clear. Article #40 is a mechanistic study. Because the sentence says that it is a well-accepted model, some more general reference would also be helpful, such as a review describing the current theory in more detail and that cites many articles. 

7.     Line 193-194. This sentence should say, “Nevertheless, multiple studies show how…”

8.     Figure 3 is very confusing to me. If there is an arrow to something, it means one thing stimulates something else, but if there is an X over that arrow, it means that something is inhibiting that stimulation. Is the X meant to be the action of insulin in all the various locations? The lines between insulin and PEPCK and GP have an inhibitory mark rather than an arrow, then there is an X over that, which should mean that Insulin stimulates PEPK and GP. I think it would be clearer to have the inhibition markings used for PEPK and GP (i.e, --|), making them bigger in each of the locations and without the X. From my understanding of the figure, I think this inhibitory marking should be between insulin and ALAS1, Insulin and PEPCK, Insulin and GP, IDE and Insulin, and ROS and IDE. Why are some of the lines dashed and others solid? This is not clear to me and should be explained if there is significance to this. FOXO1 and PGC-1a are listed, but what they are doing there is not clear. Perhaps an up arrow could be by FOXO1 and a down arrow could be by PGC-1a to indicate the change in activity of the two with increased insulin. 

9.     Lines 216-226. It seems that this who section is supposed to be the legend of Figure 3, but it is written in line with the rest of the text. The way the abbreviations are given makes this clear that this section is supposed to the legend, but it is in the wrong place. This legend text should start immediately after the Figure 3 title and be smaller text, as the legends are for the other figures. 

10.  Line 266. The authors say that they found that prophylactic administration of Ins-ApoAI and glucose was not sufficient to achieve biochemical protection against a severe attack, but a reference is not given. Please provide a reference if this is published data. If it is unpublished, please say “(unpublished data)” at the end.

11.  Line 267. The authors say that Hepatic transcriptome of AIP mice revealed that phenobarbital induced dysregulation of the gene involved in mitochondrial biogenesis and oxidative phosporphylation…,” but they do not say what that gene is. Please include the name of the gene, as it seems missing from the sentence. It should say, “ Induced dysregulation of [gene name], the gene involved in mitochondrial…”

12.  Line 268 and 277. OXPHOS is only used two times. It is better just to write out oxidative phosphorylation both times and not abbreviate it to make it easier on the reader. 

13.  Line 287-289. I am not sure what this sentence means. Please revise. I suspect the authors might be meaning the following: “Fasting secondary to nausea and vomiting and major losses of hepatic succinyl-CoA during an acute attack, as well as impaired glucose metabolism, reduces the availability of energy metabolites and could play a role in modulating the severity of porphyria attacks.”

14.  Line 291, “Intravenous” should not be capitalized. 

15.  Many patients with AIP get IV glucose with attacks, and other receive prophylactic IV glucose. Please make some mention of this and if this could be affecting the results of some of the studies described, such as reference 39. 

Author Response

Dear Reviewer,

Thank you so much for your comprehensive review. Your accurate comments have been included in the revised manuscript and we sincerely believe that with your contributions the article has improved significantly.

  1. We have modified the manuscript accordingly.
  2. We have changed the sentence according to your suggestion
  3. We have removed the unnecessary “an.”
  4. We have modified the manuscript accordingly.
  5. We have changed the sentence according to your suggestion
  6. You were right. We added one reference to support that sentence.
  7. We have changed the sentence according to your suggestion
  8. We have changed the figure 3 according to your recommendations.
  9. You were right, it was the legend of Figure 3.
  10. We have provided the reference of these data.
  11. We have included the include the name of the gene
  12. We have modified the manuscript accordingly.
  13. We have changed the sentence according to your suggestion
  14. We have changed the capital letter of intravenous
  15. We have included a paragraph in the conclusions about it.

Reviewer 2 Report

This is a good review of interesting aspects of AIP illustrating the critical influence of dietary factors and metabolism on phenotypic expression of the gene mutations, particularly wrt to insulin related metabolic function.

The authors have cited a number of experimental papers.

I wonder whether they have considered  possible relevance of 2 papers by a Japanese group using a +/- ALAS1 KO mouse.

Heterozygous disruption of ALAS1 in mice causes an accelerated age-dependent reduction in free heme, but not total heme, in skeletal muscle and liver. Archives of Biochemistry and Biophysics 697 (2021) 108721

5-Aminolevulinic acid (ALA) deficiency causes impaired glucose tolerance and insulin resistance coincident with an attenuation of mitochondrial function in aged mice. PLoS ONE (2018) 13(1): e0189593. https://doi.org/ 10.1371/journal.pone.0189593

References 28 and 53 are the same.

Lines 229-231.  Is this correct? Is it what you want to say?

This is a good review of interesting aspects of AIP illustrating the critical influence of dietary factors and metabolism on phenotypic expression of the gene mutations, particularly wrt to insulin related metabolic function.

The authors have cited a number of experimental papers.

I wonder whether they have considered  possible relevance of 2 papers by a Japanese group using a +/- ALAS1 KO mouse.

Heterozygous disruption of ALAS1 in mice causes an accelerated age-dependent reduction in free heme, but not total heme, in skeletal muscle and liver. Archives of Biochemistry and Biophysics 697 (2021) 108721

5-Aminolevulinic acid (ALA) deficiency causes impaired glucose tolerance and insulin resistance coincident with an attenuation of mitochondrial function in aged mice. PLoS ONE (2018) 13(1): e0189593.

References 28 and 53 are the same.

Lines 229-231.  Is this correct? Is it what you want to say?

Author Response

Dear Reviewer,

Thank you so much for taking the time to review our article. Your comments have been very useful.

  1. We have included the reference you suggested, including a paragraph in the conclusions
  2. We have deleted the 53 reference that was repeated. 
  3. We have changed the sentence according to your suggestion

Round 2

Reviewer 1 Report

Thank you for the revisions. The manuscript is clearer now. The is an important manuscript.